Five new species and one new genus of recent miliolid foraminifera from Raja Ampat (West Papua, Indonesia)

Förderer Meena
Langer Martin R. martin.langer@uni-bonn.de
Steinmann Institut, Paleontology, University of Bonn , Bonn , NRW , Germany
Reimer James
Electronic publication date: 2016 Jun 23
Publication date: 2016
Volume: 4
Electronic Location ID: e2157
Received 2016 Feb 27; Accepted 2016 May 31
Copyright: ©2016 Förderer and Langer
Copyright year: 2016
Copyright holder: Förderer and Langer
License: This is an open access article distributed under the terms of the Creative Commons Attribution License, which permits unrestricted use, distribution, reproduction and adaptation in any medium and for any purpose provided that it is properly attributed. For attribution, the original author(s), title, publication source (PeerJ) and either DOI or URL of the article must be cited.
License URL: https://creativecommons.org/licenses/by/4.0/

Keywords: Foraminifera, Coral Triangle, Raja Ampat, Indonesia, Tropical reefs, Benthic, Protists

Funding: German Science Foundation La-884/10-1/13-1 University of Bonn Collection and study of the material was supported by grants from the German Science Foundation (La-884/10-1/13- 1) and the University of Bonn. The funders had no role in study design, data collection and analysis, decision to publish, or preparation of the manuscript.

==============================
Raja Ampat is an archipelago of about 1,500 small islands located northwest off the Bird’s Head Peninsula of Indonesia’s West Papua province. It is part of the Coral Triangle, a region recognized as the “epicenter” of tropical marine biodiversity. In the course of a large-scale survey on shallow benthic foraminifera we have discovered one new genus and five new species of recent miliolid benthic foraminifera from the highly diverse reefal and nearshore environments. The new fischerinid genus Dentoplanispirinella is characterized by its planispiral coiling and by the presence of a simple tooth, that differentiate it from Planispirinella Wiesner. It is represented in our sample material by the new species Dentoplanispirinella occulta. The other four species described herein are Miliolinella moia, Miliolinella undina, Triloculina kawea and Siphonaperta hallocki. All new species are comparatively rare and occur sporadically in the sample material. Detailed morphological descriptions, scanning electron microscopy pictures of complete and dissected specimens as well as micro-computed tomography images are provided.

Introduction

The Raja Ampat Archipelago (West Papua, Indonesia) off the northwestern coast of New Guineas Bird’s Head Peninsula (Fig. 1B) is one of the most species rich marine environments (Erdmann & Pet, 2002; McKenna, Allen & Suryadi, 2002), situated in the Indo-Pacific’s “epicenter” of biodiversity, commonly referred to as the Coral Triangle (Hoeksema, 2007). The Coral Triangle encompasses a large part of the tropical marine waters of Indonesia, the Philippines, Malaysia, the Solomon Islands, Papua New Guinea and Timor-Leste (Fig. 1A). It includes ecoregions that are each home to at least 500 species of hermatypic corals and also show extraordinary diversity among coral associated species (Veron, 1995; Veron et al., 2009; Roberts et al., 2002; Bellwood & Hughes, 2001; Tittensor et al., 2010). The region is recognized as a “species factory” and functions as the most significant net exporter of biodiversity for adjacent reef regions (Briggs & Bowen, 2013; Ekman, 1953).

Figure 1 Maps of the sampling area.

(A) Area of the Coral Triangle (shaded) in the Central Indo-Pacific; (B) location of Raja Ampat northwest of the Bird’s Head Peninsula (West Papua, Indonesia); (C) location of sample stations where the species described herein occur (for details see Table 1).

Comprehensive studies on benthic foraminifera from the central Indo-Pacific region began with marine scientific expeditions in the late 1800s with the report on the Challenger Foraminifera by Brady (1884), and the work of Millett (1898–1904) from the Malay Archipelago. In the 20th century, systematic surveys were conducted around the Philippines (Cushman, 1921; Graham & Militante, 1959), in the Papuan Lagoon near Port Moresby, Papua New Guinea (Haig, 1988a; Haig, 1988b; Haig, 1993), in the Timor Sea and Sahul Shelf (Loeblich & Tappan, 1994), in Madang, eastern Papua New Guinea (Langer, 1992; Langer & Lipps, 2003) and more recently in the Ningaloo Reef area at Australia’s northwest coast (Parker, 2009), at Chuuk Island of the Caroline reefs (Makled & Langer, 2011) and around New Caledonia (Debenay, 2012). Recent environmental and biogeographic studies on larger benthic foraminifera in the tropical waters of the central Indo-Pacific were conducted by Langer & Hottinger (2000), Renema (2003), Renema & Hohenegger (2005), Renema (2006), Renema (2010), Renema & Troelstra (2001), Hohenegger (2004), Hohenegger (2011), Weinmann et al. (2013) and Prazeres, Uthicke & Pandolfi (2016).

To date, however, large-scale systematic studies on benthic foraminifera from Raja Ampat are lacking. The archipelago consists of the four main islands Waigeo, Batana, Salawati, and Misool, and hundreds of small satellite islets, which are largely uninhabited. Due to its remote location and difficult access conditions the coral reefs of the region remained relatively unexplored and pristine. However, increasing exposure to exploitation have required the establishment of several marine protected areas (Agostini et al., 2012). The first and to date only report on benthic foraminifera from Raja Ampat is that of Hofker (1927) and Hofker (1930), who examined the material taken by the Siboga Expedition (1899–1900) that included five samples from Raja Ampat. He documented nine species of benthic foraminifera including eight rotalid taxa and the miliolid symbiont bearing species Peneroplis pertusus (Forskål).

Material and Methods

This study was conducted with 30 sediment samples from the Raja Ampat Archipelago (New Guinea, Indonesia) from around the islands of Waigeo, Batana, Kawe, Fam and adjacent small islets in an area that covers about 2,500 km2 (Fig. 1C). The archipelago is located in the central Indo-Pacific warm pool with an average annual sea surface temperature of 29°C (Mangubhai et al., 2012). Raja Ampat is further situated in the passage way of the Indonesian Throughflow, a major ocean current that leads water masses from the western Pacific to the eastern Indian Ocean. Previous studies have shown that the reef fauna of Raja Ampat is strongly current dependent (Devantier, Turak & Allen, 2009; Turak & Souhoka, 2003).

The samples were collected by snorkeling and SCUBA diving in September 2011 by M Langer. Sediment surface samples from the top 2 cm were collected from the fore-reef slope of fringing reefs, with two samples from a patch reef, and two samples from a sandy channel with sparse coral cover (Table 1). The sediment was predominantly carbonaceous (∼90%) and included fine-grained sediments as well as coarse reef rubble. All samples were washed through a 63 µm sieve and dried at 50°C in an oven overnight. Foraminifera were picked from each sample and the best preserved specimens were imaged using a Tescan VEGA MV2300 Scanning Electron Microscope (SEM) at the Steinmann Institute of the University of Bonn. Digital plates were assembled using Adobe Photoshop CS6. Micro-computer tomography (CT) scan imaging was conducted using a phoenix v|tome|x s computed tomography system at the Steinmann Institute and visualization was carried out with Avizo 7.1.0. The new species and the new genus are described in detail using the supra-generic classification of Loeblich & Tappan (1987).

Table 1 Collection sites.

Sample site information for collection locations from Raja Ampat (Indonesia) including environmental information on reefal habitat type.

Station	Depth (m)	Latitude	Longitude	Reef- or eco-type	Reef-zone	
B14	41	0°5′22.40″N	130°13′35.29″E	Fringing reef	Fore-reef slope	
B15	43	0°5′22.40″N	130°13′35.29″E	Fringing reef	Fore-reef slope	
E23	24	0°16′25.87″S	130°18′59.19″E	Fringing reef	Fore-reef slope	
FW	49	0°35′19.86″S	130°17′45.54″E	Fringing reef	Fore-reef slope	
M21	27	0°29′50.40″S	130°43′37.62″E	Patch reef	Fore-reef slope	
MG	18	0°35′23.40″S	130°18′54.54″E	Patch reef	Platform	
MI05	32	0°16′25.87″S	130°18′59.19″E	Fringing reef	Fore-reef slope	
MI06	32	0°16′25.87″S	130°18′59.19″E	Fringing reef	Fore-reef slope	
MR17	12	0°5′47.58″S	130°14′9.66″E	Fringing reef	Fore-reef slope	
MR18	18	0°5′47.58″S	130°14′9.66″E	Fringing reef	Fore-reef slope	
MS03	16	0°34′47.88″S	130°32′32.04″E	Sand channel, sparse coral cover	Channel	
MS04	14	0°34′47.88″S	130°32′32.04″E	Sand channel, sparse coral cover	Channel	
N18	30	0°10′22.74″N	130°0′22.38″E	Fringing reef	Fore-reef slope	
U16	45	0°5′49.13″N	130°13′59.08″E	Fringing reef	Fore-reef slope	
W07	24	0°15′21.72″S	130°17′32.16″E	Fringing reef	Fore-reef slope	
Y24	26	0°47′8.64″S	130°45′25.62″E	Fringing reef	Fore-reef slope	
Y25	26	0°47′8.64″S	130°45′25.62″E	Fringing reef	Fore-reef slope	

Repository of the Material: the holotypes and topotypic paratypes of the new species are deposited in the micropaleontology collection of the Steinmann Institute of Paleontology at the University of Bonn, Germany (MaLaPNG 2011–10, MaLaPNG 2011–11, MaLaPNG 2011–12, MaLaPNG 2011–13, MaLaPNG 2011–14).

The electronic version of this article in Portable Document Format (PDF) will represent a published work according to the International Commission on Zoological Nomenclature (ICZN), and hence the new names contained in the electronic version are effectively published under that Code from the electronic edition alone. This published work and the nomenclatural acts it contains have been registered in ZooBank, the online registration system for the ICZN. The ZooBank LSIDs (Life Science Identifiers) can be resolved and the associated information viewed through any standard web browser by appending the LSID to the prefix http://zoobank.org/. The LSID for this publication is: urn:lsid:zoobank.org: pub:FB001C3C-AEA9-45D5-9224-EDD084378897. The online version of this work is archived and available from the following digital repositories: PeerJ, PubMed Central and CLOCKSS.

Results

Smaller miliolid benthic foraminifera are typical dwellers in surface sediments of shallow water reefal and lagoonal habitats. By studying the highly diverse assemblages of benthic foraminifera from Raja Ampat, taken from different locations around the islands (Fig. 1C), we recorded a total of 455 species among them 249 miliolid species, of which five are described here as new. Four species belong to the widely distributed miliolid genera of Miliolinella Wiesner, Triloculina d’Orbigny and Siphonaperta Vella. As the morphological properties of the fifth species differentiate it from any previously known genera, we designate and describe it as the new genus Dentoplanispirinella.

Systematic Descriptions

Subclass Miliolana Saidova, 1981	
Order Miliolida Delage & Hérouard, 1896	
Suborder Miliolina Delage & Hérouard, 1896	
Superfamily Cornuspiracea Schultze, 1854	
Family Fischerinidae Millett, 1898	
Subfamily Fischerininae Millett, 1898	

Figure 2 Holotype, paratype, CT scans and details of Dentoplanispirinella gen. nov. occulta sp. nov.

(A) Side view and (B) apertural view of a more juvenile specimen with a nearly triangular aperture and weakly developed tooth (paratype); (C) side view and (D) apertural view (holotype); (E) apertural view of a specimen with a well-developed tooth and elongated aperture; (F) detail of a well-developed peripheral keel; (G) CT scan reconstruction of the chamber cavities revealing the presence of 2.5 – 3.5 chambers per whorl in an adult specimen (note that penultimate chamber is broken); (H) CT scan showing planispirally arranged chambers; (I) detail of an aperture with a very well-developed tooth; (J) detail of the striate surface ornamentation; (K) detail of the construction of the outer wall layer showing randomly arranged calcite needles in the lower part (test surface removed) and longitudinally arranged calcite needles on the outer test surface. Scale bar is 100 µm (unless indicated).

Genus Dentoplanispirinella Förderer and Langer gen. nov.

urn:lsid:zoobank.org:act:98A1DD41-C0AE-4401-830B-0D189E70661A

Description. Test small, broadly circular in outline, discoidal to slightly biconvex. Periphery with a weakly developed subrounded keel that encircles the entire test margin. Wall thick, calcareous, porcelaneous, imperforate. Coiling involute, throughout planispirally enrolled with 2.5 to 3.5 tubular chambers per whorl, each whorl slightly offset to the proceeding coil with a tendency to become sigmoiline (axial section as seen in CT scan, Fig. 2H). Lateral wall extensions of the adult chambers entirely cover the earliest chambers and tend to overlap the umbilical region in each whorl. Sutures oblique, thin and irregular. Aperture arch-shaped, triangular in juvenile specimens, high and subtriangular in adult specimens, tapering apically, on the base connected with the peripheral margin of the proceeding chamber and provided with a very small and thin tooth. In juvenile specimens the tooth appears just like a little knob or slightly raised spine.

Type species. Dentoplanispirinella occulta sp. nov.

Remarks. Dentoplanispirinella gen. nov. resembles Planispirinella Wiesner, 1931 in having a discoidal shape, a high aperture and a planispiral chamber arrangement, but differs from Planispirinella by the presence of a tooth and the more biconvex test shape in apical view (Fig. 2B). The apertural features and the coiling mode of Dentoplanispirinella further distinguish it from Nummoloculina Steinmann 1881, which has an apertural flap and an early quinqueloculine coiling.

Dentoplanispirinella occulta Förderer and Langer sp. nov.	
Figures 2A–2K	
urn:lsid:zoobank.org:act:7E132939-5284-484D-9B50-BC79A0B52D0A	

Etymology. From the Latin “occultare” meaning for “hiding.”

Material. 28 specimens from nine samples (MR18, MI05, MI06, MS03, MS04, MG, M21, U16, Y24; Fig. 1C; Table 1), recent.

Holotype. The specimen illustrated here as Figs. 2C and 2D (sample MS03; MaLaPNG 2011–10).

Paratype. The specimen illustrated here as Figs. 2A and 2B (sample MS03; MaLaPNG 2011–10).

Type locality. The holotype and the paratype are from sample station MS03 (16m), a sand channel between Arborek Island and Pulau Mansuar; Raja Ampat, New Guinea (Indonesia).

Diagnosis. A species of Dentoplanispirinella gen. nov. with a discoidal to biconvex test shape, a slightly keeled periphery, a radial oriented, finely striate surface ornamentation and an arch-shaped, triangular aperture, provided with a small tooth.

Dimensions. Observed range of test dimensions: diameter 285–704 µm (lateral view), test width 100–193 µm (apertural view).

Occurence. Dentoplanispirinella occulta is widely distributed in the Raja Ampat area in fine to coarse coral rubble samples from depths of 14 to 45 m.

Description. Test porcelaneous and imperforate. Almost circular in lateral view, lenticular and biconvex in apertural view with a slightly developed, subrounded keel and weakly inflated chambers. Coiling planispiral and involute. Two and a half to three and a half chambers visible from the exterior. Lateral wall extensions of the adult chambers entirely cover the earliest chambers and tend to overlap the umbilical region; the final chamber covers approximately half of the test surface. Sutures oblique, thin, irregular and recurved near the periphery. Test surface ornamented with radial oriented, fine, sub-parallel to anastomosing striae that are straight to slightly curved backwards, towards the outer margins of the chambers. Umbilical region and test periphery more weakly ornamented. Outer wall layer constructed of longitudinally aligned needle-shape crystals, oriented perpendicular to direction of ornamentation. The test appears matte white under the light microscope with a slightly translucent periphery. Apertural face not ornamented. Aperture arch-shaped and triangular in juvenile specimens, high and subtriangular in adult specimens, tapering apically, on the base connected with the peripheral margin to the preceding chamber and provided with a peristomal rim. Aperture provided with a very small and thin tooth, with the flat side oriented in lateral direction.

Remarks. Dentoplanispirinella occulta sp. nov. differs from Planispirinella involuta Collins (1958, p. 374, pl. 4, Figs. 2A and 2B) by its more lenticular biconvex shape in horizontal section, the subtriangular shape of the aperture, the presence of a small tooth, and the striate surface ornamentation.

Superfamily Miliolacea Ehrenberg, 1839	
Family Hauerinidae Schwager, 1876	
Subfamily Hauerininae Schwager, 1876	

Genus Miliolinella (Wiesner, 1931)

Miliolinella moia Förderer and Langer sp. nov.	
Figures 3A–3L	
urn:lsid:zoobank.org:act:D8184E0C-2805-40D7-BCCB-492D74216168	

Etymology. The new species is named after the indigeneous Moi people from Malaumkarta, a Papuan tribe from the north coast near Sorong.

Figure 3 Holotype and paratypes of Miliolinella moia sp. nov.

(A–C) Holotype, five chambers visible from the exterior: (A) lateral view of more evolute side; (B) top view; (C) lateral view of more involute side; (D–F) paratype, a specimen with a broken ultimate chamber showing three chambers visible from the exterior: (D) lateral view of more evolute side; (E) top view; (F) lateral view of more involute side; (G–I) a specimen with four chambers visible from the exterior: (G) lateral view of more evolute side; (H) top view; (I) lateral view of more involute side; (J–L) a specimen with four chambers visible from the exterior: (J) lateral view of more involute side; (K) top view; (L) lateral view of more evolute side. Scale bar is 100 µm.

Material. 11 specimens from six samples (B14, B15, E23, MR17, N18, U16; Fig. 1C; Table 1), recent.

Holotype. The specimen illustrated here as Figs. 3A–3C (sample B14; MaLaPNG 2011–11).

Paratypes. The specimens illustrated here as Figs. 3D–3F (sample B14), Figs. 3G–3I and Figs. 3J–3J (sample ER23; MaLaPNG 2011–11).

Type locality. The holotype and the paratype are from sample station B14 (41 m), Bag Island, east of Pulau Uranie; Raja Ampat, New Guinea (Indonesia).

Diagnosis. A slightly enlongated, medium-sized species of Miliolinella Wiesner, 1931 with a compressed, angular and slightly slanted outline, a smooth and shiny wall, and a high subcircular opening.

Dimensions. Observed range of test dimensions: test height 409–554 µm, test width 278–396 µm (lateral view), 166–250 µm (apertural view).

Occurence. This species is widely distributed in the Raja Ampat area in fine to coarse coral rubble samples and occurs at depths between 12 and 45 m.

Description. Test porcelaneous and imperforate, ovate in outline and slightly higher than broad. Test weakly compressed and flattened, subtriangular in apertural view. Chamber arrangement quinqueloculine with five chambers visible from the exterior. In some specimens only three to four chambers are visible. Periphery rounded to subrounded, chambers slightly inflated. Sutures curved, distinct and weakly depressed. Chambers tend to be off-centered, giving them a slanted appearance. Test wall smooth, translucent to opaque and glossy under the light microscope. Aboral end of the chambers slightly constricted. Aperture in basal position, a Miliolinella-type large subcircular opening with an everted peristomal rim and a semicircular, slightly excavated flap, that covers more than half of the opening.

Remarks. Miliolinella moia sp. nov. differs from Miliolinella pilasensis McCulloch, 1977 (p. 566, pl. 238, Fig. 16 and Loeblich & Tappan, 1994, p. 57, pl. 99, Figs. 1–9) in its angular and more compressed outline, and the large subcircular opening. Millet (1898) depicted a species of Miliolina valvularis (Reuss) from the Malay Archipelago (p. 11, Figs. 5A–5C) that shows a high degree of similarity to Miliolinella moia, but his specimen has a more rounded periphery. The original description of Triloculina valvularis by Reuss (1851, p. 85, pl. 7, Fig. 56) shows a specimen with a broadly rounded periphery and inflated chambers without angles. Miliolinella sp. 2 figured in Parker, 2009 from Ningaloo Reef, Australia (p. 128, Figs. 92A–92I, 93A–93J, 94A–94K) differs from Miliolinella moia by the low apertural opening and the broadly rounded and more inflated chambers.

Figure 4 Holotype and paratypes of Miliolinella undina sp. nov.

(A–C) Holotype: (A) oblique apertural view; (B) apertural view; (C) lateral view of more involute side; (D–F) a specimen with the final chamber missing: (D) lateral view of more evolute side; (E) top view; (F) lateral view of more involute side; (G–I) a specimen with an erratic growth stage in the final chambers: (G) side view; (H) top view; (I) lateral view of more involute side. Scale bar is 50 µm.

Miliolinella undina Förderer and Langer sp. nov.	
Figure 4A–4I	
urn:lsid:zoobank.org:act:D11E1426-9DCC-41B8-A992-27D974A92520	

1988a Miliolinella sp. B—Haig, Papuan Lagoon, Port Moresby, p. 224, pl. 2, Figs. 23 and 24.	
1992 Miliolinella sp.—Hatta & Ujiié, Ryukyu Islands, p. 72, pl. 10, Fig. 6.	
?2012 Miliolinella cf. M. semicostata (Wiesner)—Debenay, New Caledonia, p. 110, 275.	

Etymology. After the undulate ornamentation of the test. From the Latin “unda” meaning wave and mythological “Undine,” a term established by the Renaissance alchemist Paracelcus for water spirits.

Material. Three specimens from three samples (MR18, N18, U16; Fig. 1C; Table 1), recent.

Holotype. The specimen illustrated here as Figs. 4A–4C (sample MR18; MaLaPNG 2011–12).

Paratypes. The specimens illustrated here as Figs. 4D–4F (sample N18) and Figs. 4G–4I (sample U16; MaLaPNG 2011–12).

Type locality. The holotype is from sample station MR18 (18 m), east of Kawe Island. The paratypes are from sample stations N18 (30 m), south-west coast of Pulau Wayag, and U16 (45 m), between Pulau Uranie and Bag Island; Raja Ampat, New Guinea (Indonesia).

Diagnosis. A small quinqueloculine species of Miliolinella Wiesner with inflated chambers, a rounded outline and an undulate to reticulate surface ornamentation.

Description. Test porcelaneous and imperforate, small, ratio of height and width variable but usually slightly higher than broad. Periphery rounded and chambers slightly inflated. Chamber arrangement quinqueloculine, with five chambers visible from the exterior. Aboral end rounded, flush with the surface in the holotype to slightly raised in paratypes. Wall smoothly finished, matte, translucent under the light microscope. Sutures curved and depressed. Test surface ornamented with numerous irregular, predominantly longitudinal, somehow honeycombed reticulate to undulate low anastomosing costae that are covering large parts of the test. Outer-wall layer constructed of needle-shaped crystals that are primarily aligned in longitudinal direction. Aperture basal, a large semicircular Miliolinella-type opening, provided with a thickened and everted peristomal rim and a broad, slightly excavated basal flap.

Dimensions. Observed range of test dimensions: test height 146–162 µm, test width 114–224 µm (lateral view), 81–119 µm (apertural view).

Occurence. Miliolinella undina is present with one specimen in each of three highly diverse, miliolid-rich, fine coral rubble samples from depths of 18 to 45 m.

Remarks. Specimens of Miliolinella undina sp. nov. have been previously documented by Haig, 1988a as Miliolinella sp. B from the Papuan Lagoon, Port Moresby and by Hatta & Ujiié, 1992 as Miliolinella sp. from the Ryukyus. Hatta & Ujiié mentioned the species to occur rarely in their assemblages. The new species has also been recorded in samples from northern Palawan (M Förderer, 2016, unpublished data). Miliolinella cf. M. semicostata (Wiesner) depicted by Debenay from New Caledonia (2012, p. 110, 275) may also belong to Miliolinella undina, but shows a less undulated test ornamentation. Test shape, apertural and ornamental features are more similar to our holotype (Figs. 4A–4C) than to Miliolinella semicostata (Wiesner, 1923) from the Mediterranean Sea (see Cimerman & Langer, 1991, p. 42, pl. 38, Figs. 10–15). Miliolinella semicostata has less inflated chambers and the ornamentation is not reticulate but longitudinally striate and restricted to the angles. Miliolinella undina also resembles Miliolinella sp. 4 depicted by Parker from the Ningaloo Reef in Western Australia (2009, p. 136, Figs. 97A–97H), but his specimen has a less undulated and more striate alignment of costae. The new species also resembles Miliolinella flintiana (Cushman, 1932) (p. 55, pl. 12: 4A–4C) in size, test shape, chamber arrangement and apertural features. However it differs in its surface ornamentation, that is distinctly longitudinal costate in Miliolinella flintiana and undulate and more irregular in Miliolinella undina. Miliolinella flintiana also occurs in our assemblages.

Genus Triloculina D’Orbigny, 1826

Triloculina kawea Förderer and Langer sp. nov.	
Figs. 5A–5H	
urn:lsid:zoobank.org:act:6F5B38CE-88B3-4FBE-9329-8483756158E1	

2009 Triloculina? sp. 2—Parker, Ningaloo Reef, p. 372, Figs. 271F–271J.

Etymology. This species is named in honor of the indigeneous people of West Papua after the Kawe tribe, that owns and protects a highly diverse marine protected area of Raja Ampat.

Material. 12 specimens from seven samples (B15, FW, M05, MS04, N18, U16, Y25; Fig. 1C; Table 1), recent.

Holotype. The specimen illustrated here as Figs. 5A–5C (sample FW; MaLaPNG 2011–13).

Paratype. The specimen illustrated here as Figs. 5E–5G (sample FW; MaLaPNG 2011–13).

Figure 5 Holotype, paratype, cross section and detail of Triloculina kawea sp. nov.

(A–C) Holotype: (A) lateral view of more involute side; (B) apertural view; (C) lateral view of more evolute side; (D) cross section of a specimen; (E–G) Paratype: (E) lateral view of more evolute side; (F) apertural view; (G) lateral view of more involute side; (H) detail of the irregular test surface. Scale bar is 100 µm (unless indicated).

Type locality. The holotype and the paratype are from sample station FW (49 m), south-east Penemu, Fam Islands; Raja Ampat, New Guinea (Indonesia).

Diagnosis. A medium-sized species of Triloculina d’Orbigny with a slightly elevated “Lachlanella”-type aperture, rounded periphery, blunt angles and a roughly textured wall.

Dimensions. Observed range of test dimensions: test height 377–439 µm, test width 200–245 µm (lateral view), 162–195 µm (apertural view).

Occurence. This species is widely distributed in our sampling area in fine to coarse coral rubble samples from depths of 14 to 49 m.

Description. Test porcelaneous and imperforate, about one and a half times longer than broad. Broadly triangular in apertural view, ovate in outline. Chamber arrangement triloculine, periphery rounded to subrounded, chambers inflated with blunt angles. Sutures distinct and depressed. Surface ornamented with elongated, irregular longitudinal arranged short striae covering the entire test surface, giving the appearance of a matte and roughly textured wall under the light microscope. Outer wall layer consisting of longitudinally aligned plate shaped crystals. Aboral end rounded and slightly produced, oral end produced and connected with the peripheral margin of the preceeding chamber. Aperture basal, “Lachlanella”-type with a long slender tooth that becomes thickened at the tip.

Remarks. The species Triloculina? sp. 2 reported by Parker, 2009 from Western Australia differs from Triloculina kawea sp. nov. in its less triangular shape and less elongated outline. We consider Parker’s specimen a juvenile individual of Triloculina kawea. The aperture of Parker’s specimen is not intact but resembles very well the apertural features of Triloculina kawea. The outer wall layer appears identical (Fig. 5H). Parker mentioned the species to be possibly cryptoquinqueloculine. Figures 5B and 5F and the horizontal section (5D) show the triloculine chamber arrangement. Triloculina sp. 1, reported by Debenay 2012 from New Caledonia (p. 139, 278) is very similar in shape and surface ornamentation to Triloculina kawea, but has significantly more acute angles and a Y-shaped tooth. Triloculina kawea further differs from Triloculina linneiana d’Orbigny depicted by Baccaert, 1987 from the Great Barrier Reef (p. 128, pl. 57, Figs. 3 and 4) in the less striate ornamentation and more acute angles.

Subfamily Siphonapertinae Saidova, 1975

Genus Siphonaperta Vella, 1957	
Siphonaperta hallocki Förderer and Langer sp. nov.	
Figs. 6A–6F	
urn:lsid:zoobank.org:act:DD4F0DB3-1355-4BB1-841A-FFE32E0F6455	

?1988a Quinqueloculina sp. C—Haig, Papuan Lagoon, Port Moresby, p. 234, pl. 9, Figs. 7–10.	
?2009 Quinqueloculina sp. 13—Parker, Ningaloo Reef, p. 311, Figs. 224A–224J, 225A–225G.	

Etymology. In honor of Pamela Hallock Muller for her extensive work on tropical foraminifera.

Figure 6 Holotype and paratype of Siphonaperta hallocki sp. nov.

(A–C) Holotype: (A) lateral view of more evolute side; (B) apertural view; (C) lateral view of more involute side; (D–F) paratype: (D) lateral view of more evolute side; (E) apertural view; (F) lateral view of more involute side. Scale bar is 100 µm (unless indicated).

Material. Four specimens from three samples (MS03, N18, W07; Fig. 1C; Table 1), recent.

Holotype. The specimen illustrated here as Figs. 6A–6C (sample MS03; MaLaPNG 2011–14).

Paratype. The specimen illustrated here as Figs. 6D–6F (sample N18; MaLaPNG 2011–14).

Type locality. The holotype is from sample station MS03 (18 m), a sand channel between Arborek Island and Pulau Mansuar. The paratype is from sample station N18 (30 m), south-west coast of Pulau Wayag; Raja Ampat, New Guinea (Indonesia).

Diagnosis. A medium-sized species of Siphonaperta Vella with a finely agglutinated wall, carinate shoulders, a short neck and a circular aperture with a small bifid tooth.

Description. Test porcelaneous and imperforate, medium-sized, about two times longer than broad, and ovate in outline. Outer layer of the calcareous test wall covered with finely agglutinated mostly biogenic grains. Agglutinated grains are particularly frequent along the sutures. Periphery carinate to subacute. Chamber arrangement quinqueloculine with five chambers visible from the exterior. Sutures slightly curved, incised and depressed. Chambers weakly inflated and angular in section, with weakly developed longitudinal striae (in well preserved specimens). Aboral end rounded and produced; oral end becoming more slender and leading into a short produced neck. Aperture terminal, a wide circular opening with a short T-shaped, bifid tooth, that reaches more than one third of the apertural diameter. Apertural opening surrounded by a slightly thickened and everted peristomal rim.

Dimensions. Observed range of test dimensions: test height 240–442 µm, test width 132–233 µm (lateral view), 87–119 µm (apertural view).

Occurence. Siphonaperta hallocki occurs sporadically in fine to coarse coral rubble samples from depths of 16 to 30 m.

Remarks. Very similar specimens were previously documented as Quinqueloculina sp. C from the Papuan Lagoon (Haig, 1988a) and Quinqueloculina sp. 13 from Ningaloo Reef (Parker, 2009). Test shape, wall texture and apertural features appear to be identical to our specimens from Raja Ampat. Quinqueloculina sp. 4 documented by Parker, 2009 from the Ningaloo Reef appears very similar to Siphonaperta hallocki, but differs in its more elongated shape, more rounded and inflated chambers and the cryptoquinqueloculine coiling. In addition, Parker describes the wall as roughly textured with some agglutinated grains. Quinqueloculina tropicalis Cushman from Samoa (1924, p. 63, pl. 23, Figs. 9 and 10) differs from our new species by its more compressed shape and more elongated broadly rounded chambers without any angles or costae. Quinqueloculina polygona D’Orbigny (1839, p. 198,pl. 12, Figs. 21–23) differs from Siphonaperta hallocki in its smooth and shiny surface, the pronounced carinae and the less inflated chambers. Langer et al., 2013 depicted a specimen of Quinqueloculina polygona d’Orbigny from Bazaruto (Langer et al., 2013, p. 163, Fig. 5: 14) that resembles our new species in size, shape and apertural features. However, it is unlikely that this species from Bazaruto belongs to Siphonaperta hallocki, as its outer wall layer is not agglutinated.

The authors gratefully acknowledge Dr. Stephanie Pietsch for assistance with the collection of the samples, Georg Oleschinski for help with the SEM and Peter Göddertz and Kai Jäger for support with CT scan imaging. We thank Justin H. Parker, Tomas Cedhagen and an anonymous reviewer for constructive and helpful comments on the manuscript.

Additional Information and Declarations

Competing Interests

Author Contributions

Data Availability

New Species Registration

The authors declare there are no competing interests.

Meena Förderer performed the experiments, analyzed the data, contributed reagents/materials/analysis tools, wrote the paper, prepared figures and/or tables, reviewed drafts of the paper.

Martin R. Langer conceived and designed the experiments, performed the experiments, analyzed the data, contributed reagents/materials/analysis tools, wrote the paper, reviewed drafts of the paper.

The following information was supplied regarding data availability:

1) Micropaleontological Collection Steinmann Institute of Paleontology at the University of Bonn, Germany

2) MaLaPNG 2011-10, MaLaPNG 2011-11, MaLaPNG 2011-12, MaLaPNG 2011-13, MaLaPNG 2011-14.

The following information was supplied regarding the registration of a newly described species:

Dentoplanispirinella gen. nov. urn:lsid:zoobank.org:act:98A1DD41-C0AE-4401-830B-0D189E70661A

Dentoplanispirinella occulta sp. nov.

urn:lsid:zoobank.org:act:7E132939-5284-484D-9B50-BC79A0B52D0A

Miliolinella moia sp. nov.

urn:lsid:zoobank.org:act:D8184E0C-2805-40D7-BCCB-492D74216168

Miliolinella undina sp. nov.

urn:lsid:zoobank.org:act:D11E1426-9DCC-41B8-A992-27D974A92520

Triloculina kawea sp. nov.

urn:lsid:zoobank.org:act:6F5B38CE-88B3-4FBE-9329-8483756158E1

Siphonaperta hallocki sp. nov.

urn:lsid:zoobank.org:act:DD4F0DB3-1355-4BB1-841A-FFE32E0F6455

The LSID for this publication is: urn:lsid:zoobank.org:pub:FB001C3C-AEA9-45D5-9224-EDD084378897.

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
