# Peer review of "Five new species and one new genus of recent miliolid foraminifera from Raja Ampat (West Papua, Indonesia)"

_PeerJ, doi:10.7717/peerj.2157_

## Round 0.1 · original submission · Minor Revisions

· Academic Editor

Minor Revisions

Three reviewers have returned their comments to me; all were positive but offered helpful and constructive areas of improvement. Please look over all of these comments carefully. Based on these assessments, my decision is 'minor revision'.

·

Basic reporting

No Comments

Experimental design

No Comments

Validity of the findings

No Comments

Additional comments

The manuscript proposes a new genus and several new species from an area considered to be a hot spot of biodiversity and a potential species factory. So this is an important and interesting area of study, and is necessary to understand the biodiversity of the broader Indo-Pacific region and distribution of the faunas. The new genus and new species are well described and illustrated, and the microtomography images of the genotype are a very useful.
However there are several, essentially minor criticisms that need to be addressed before the manuscript is acceptable for publication:
1. The genus name Planispirinoides is preoccupied by Planispirinoides Parr 1950, and as such is unavailable. Unless the authors can demonstrate an error in Parr’s work that would make his name invalid, and thus available a new name should be proposed. The unusual radial ornament and the thickened structure of the aperture in the genotype, appears very similar to juvenile specimens of Sigmoihauerina and Pseudohauerina, where the aperture becomes eventually becomes a trematophore in the adult stage. However, in your genus the early stage is planispiral not milioline as in the above genera. It would be interesting to know if there were any larger specimens in the same samples that this species occurs in that have a trematophore and radial ornament.
2. I recommend the authors consult with the “Illustrated glossary of terms used in foraminiferal research” (available online at http://paleopolis.rediris.es/cg/CG2006_M02/) to maintain consistency in the use of foraminiferal terminology, I have made some suggestions in this regard in the annotated PDF.
3. There are several minor typographical mistakes; I recommend the authors run a spell checker before resubmitting.
4. The authors have only few specimens for some of the new species that they are establishing, and although technically only a single specimen (the holotype) is needed to erect a new species, it is impossible to capture the range of variability. This is evident in some of the specimens described as being described as “juvenile”. Comments to this regard are provided in the manuscript. It would be good if selective picks were performed on the samples to increase the number of individuals that constitute the type specimens. Keep in mind that the paratypes don’t actually have to be figured, just the specimens deposited in a repository and the respective accession numbers reported.
Additional suggestions are provided as annotations in the pdf version of the manuscript attached. The species described within, to my knowledge, have not been formally identified. Making these available will assist in future researches on foraminifera from the Indo-Pacific region. I recommend the manuscript be accepted after minor revisions as outlined in the pdf. The manuscript should not need further review upon resubmission.

Best regards,

Justin Parker

Reviewer 2 ·

Basic reporting

No Comments

Experimental design

No Comments

Validity of the findings

According to the World Modern Foraminifera Database, The new genus name "Planispirinoides" has already been named by Parr (1950).
You should reconsider the name of the new genus.

Most of new species found in this study are rare in the study area.
You should verify if the specimens could not be explained by the variations of the common nearest species together with their plates.

Additional comments

Some citations are missing. Better to recheck citations.

·

Basic reporting

No Comments

Experimental design

No Comments

Validity of the findings

No Comments

Additional comments

This manuscript is relevant and well-written and the quality of the figures is excellent. I recommend it to be published. I have only a few small comments:

The word “epicenter” is used in the abstract (line 14) and in the Introduction (line 29). This is a term used in seismology (origin of an earthquake) and should be avoided here. I suggest the word “center”.

The word “what” (line 17) would be correct in German, but “that” is probably better here.

Line 38: replace starter with started.

Line 42: The references by Haig should be written as (Haig 1988a, b, 1993) or similar.

Line 111: Check Miliolana Saidova, 1981. I think the spelling should be Miliolicea for the class and Miliolicae for the subclass.

Lines 127 and 173: consider if apical should be apically.

Line 174: delete “provided”.

Line 240: I think it is a good idea to make it clear that you [probably] mean the central European Undine (e.g., Friedrich Heinrich Karl, Freiherr de La Motte-Fouqué) and not a Papuan water spirit.

Line 270: … between … to … should be either between 18 and 45 OR: from 18 to 45.

Line 521: This refers to a series of 17 articles and I think it is important to give volume and pages to each of the articles after the journal name. Alternatively could you refer to the book published by Antiquariaat Junk in 1970, where all articles + plates are reprinted. (It also contains a new introduction and an index to all names).
Millett, F.W. 1970. Report on the Recent Foraminifera of the Malay Archipelago collected by Mr. A. Durrand, F.R.M.S., Parts 1-17. Reprinted from Journal of the Royal Microscopical Society, 1898-1904. With a New introduction by J. Hofker. Antiquariaat Junk, Lochem, 248 pp.

Legend to Fig. 1: delete “Location”.

(It is a pity that no genetic information was included. Please, try to do that in the future).

---

## Round 0.2 · Minor Revisions

· Academic Editor

Minor Revisions

Your resubmitted version was reviewed by one reviewer, who was pleased with your revisions, and I feel this paper is almost ready for publication.

Before final acceptance, however, there are some small English brush-ups that are needed, so please see my attached pdf file with comments, and go over the manuscript carefully one last time to pick up mistakes. For example, line 399 says "und" instead of "and". I have also added a few small comments here and there. Please take a look and revise accordingly.

·

Basic reporting

No Comments

Experimental design

No Comments

Validity of the findings

No Comments

Additional comments

There are still some minor English mistakes, such as "shape und apertural features" in the remarks on Siphonaperta hallocki, so I recommend another proof read by the editor before publication.

---

## Round 0.3 · accepted · Accept

· Academic Editor

Accept

Thank you for the small but important edits of the final round of revisions. I look forward to seeing this manuscript in published form!